# The Northern Stronghold Sacrifice and the Political Legitimacy of Ethnic Minority Regimes in the Late Imperial China

**Chenxi Huang [1,\*] and Siyu Chen [2]**

1  College of Philosophy, Anhui University, Hefei 230039, China
2  Harvard Divinity School, Cambridge, MA 02138, USA; schen.ucsb@gmail.com
\*  Correspondence: chenxihuang@ahu.edu.cn

**Abstract:** Traditional Chinese state sacrificial ritual represented a symbolic system of integrating religious belief, divine authority, and political legitimacy. The Northern Stronghold (Beizhen 北鎮, i.e., Mount Yiwulü 醫巫閭山) was equal in status to the other four strongholds, which, moreover, served as a strategic military fortress and represented the earth virtue in the early state sacrifice system. In the late imperial era of China, and during the Yuan (1279–1368) and Qing (1644–1911) dynasties in particular, the Northern Stronghold swiftly achieved prominence and eventually became an instrument used by minority ethnic groups, namely the Mongolians and Manchus, when elaborating upon the legitimacy of their political regimes. During the Yuan dynasty, the mountain spirits of the five strongholds (*Wuzhen* 五鎮) were formally invested as kings and, as a result, were accorded equivalent sacrifices in comparison to those given to the five sacred peaks (*Wuyue* 五嶽). Given that the Northern Stronghold was located near the northeast of Beijing, the Yuan government considered it the foundation of the state. Thereafter, the Northern Stronghold was regarded as the most important of the five stronghold mountains. In the Ming dynasty (1368–1644), the Northern Stronghold Temple (Beizhenmiao 北鎮廟) was reconstructed as both a military fortress and religious site, while its representation as a significant site for a foreign conquest dynasty diminished and its significance as a bastion of anti-insurgent suppression emerged. By the Qing dynasty, the Northern Stronghold was regarded as an integral component of the geographic origin of the Manchu people and thereby assumed once again a position of substantial political significance. Several Qing emperors visited the Northern Stronghold and left poems and prose written in graceful Chinese to present their high respect and their mastery of Chinese culture. The history of the Northern Stronghold demonstrates how the ethnic minority regimes successfully utilized the traditional Chinese state sacrificial ritual to serve their political purpose.

**Keywords:** Mount Yiwulü; Northern Stronghold; Beizhen; state sacrificial ritual; ethnic minority in northern China; legitimacy of political regime

## 1. Introduction

Stronghold mountain (*zhenshan* 镇山) sacrifice was an integral part of the traditional Chinese state ritual system of sacrifice to mountain and water spirits, which included the five sacred peaks, five strongholds, four seas (*sihai* 四海), and four waterways (*sidu* 四瀆). The earliest historical records of the "strongholds" date from the late Warring State period (403 BCE–221 BCE) to the early Han dynasty (202 BCE–220 CE) in *the Rites of Zhou* (*Zhouli* 周禮), which documented nine strongholds in nine precincts (*jiuzhou jiushanzhen* 九州九山鎮) (Zheng and Jia 2000, 33.1020–34) and four strongholds (*sizhen* 四鎮) (Zheng and Jia 2000, 22.697–98). According to Zheng Xuan's commentary, the "nine strongholds" are divided into five sacred peaks and four strongholds. The four strongholds, namely Mount Guiji 會稽山, Mount Yi 沂山, Mount Yiwulü 醫巫閭山, and Mount Huo 霍山, were the foremost mountains in their respective administrative regions like the five sacred peaks in theirs (Zheng and Jia 2000, 22.697–98, 33.1020–34). From the Han dynasty to the Northern Song,

*Zhouli*'s four strongholds were gradually added Mount Wu 吳山 to form five strongholds (Jia 2021).

According to the traditional interpretation, a "stronghold" not only refers to a great mountain but also serves to safeguard and bring stability to its nearby region (Zheng and Jia 2000, 33.1022; Wang 2019, 1a.23). Thus, the strongholds have a further military connotation than the five sacred peaks, in addition to the political and religious significance. Besides, since the Northern Stronghold (Beizhen 北鎮, i.e., Mount Yiwulü) is located in the northeastern region, where multiple ethnic minorities resided, the area had been ruled by ethnic minority regimes, such as Liao (907–1125), Jin (1115–1234), Yuan (1271–1368), and Qing (1636–1912). The Northern Stronghold was especially revered by the people of these regimes because they regarded this sacred mountain as the birthplace of their nationalities. On the other hand, because of Mount Yiwulü's frontier location, it had also been on the frontline of frequent military confrontations between the Han Chinese regimes and the minority regimes. Differences in the attitudes of the Chinese regime and ethnic minority regimes toward the northern stronghold, regardless of their similar reverence for the area, had developed because of various political, social, military, and religious reasons. While China preferred to regard the northern stronghold as a military fortress with a divine character, the minorities treated it as a source of political legitimacy for their regimes.

Previous scholarships on the Northern Stronghold have mainly involved the archaeological excavation of tombs and relics, the study of the Northern Stronghold history, and the analysis of specific stele inscriptions (Wang 2018, pp. 173–76; Wang 2019, pp. 661–69). In terms of archaeological research, studies mainly focus on two areas: The excavation of Liao-dynasty imperial tombs (Liaoning Sheng Wenwu Kaogu Yanjiusuo 2016, pp. 34–54; Yu and Bai 2020, pp. 27–33; Si et al. 2021, pp. 50–62) and the Northern Stronghold Temple architecture (Zheng 1994, pp. 42–44; Zheng et al. 1995, pp. 15–17, 27; Jia 2008, pp. 95–96; Yu 2011, pp. 235–36; Sun 2018a, pp. 143–46, 154). The research on the history of the Northern Stronghold includes organization of the sacrificial ritual (Liu 2019, pp. 34–38; Chen 2018, pp. 147–49), discussions of the ethnic minorities' practice of sacrifice (Cui 2015, pp. 112–19), and studies of the culture and palace of the Northern Stronghold in Qing Dynasty (Sun 2018b, pp. 8–10, 62; Lu 1994, pp. 71–74; Li 2002, pp. 46–48). However, little academic research has been done in Chinese on the significant implications and the political purpose of the Northern Stronghold sacrifice, and its related scholarly work in English is almost non-existent.

Although a few scholars have paid attention to the sacrificial ritual of the minority regimes in the Northern Stronghold, the precious stele inscriptions preserved in the Northern Stronghold Temple have not been fully studied. A systematic investigation of the stele inscriptions from the Yuan, Ming, and Qing dynasties in the Northern Stronghold Temple helps to trace the causes, manifestations, and evolution of the different attitudes between Han and minority governments towards the Northern Stronghold, which contributes to reevaluating the position of the religious, military, and political status of the Northern Stronghold in Chinese history. This paper examines the developmental history of the sacrifice ritual to the northern stronghold based on historical documents and extant stele inscriptions, aiming to present a historical overview that sheds light on the changing interpretations of the northern stronghold in the state ritual system of sacrifice, particularly in Yuan, Ming, and Qing Dynasties.

## 2. The Early History of the Sacrifice to the Northern Stronghold

Mount Yiwulü 醫巫閭山 is also called and written as Wulü 無慮, Yuweilü 于微閭, and lü 閭 with different Chinese characters and similar pronunciations. While Duan Yucai 段玉裁 (1735–1815) believed that the name "Yiwulü" was a transliteration of the name in the Eastern Barbarian (Dongyi 東夷) language (Xu and Duan 1988, p. 11), the explanatory sources of the derivation and specific connotations the names carry are yet to be found. Although *the Rites of Zhou* mentions Mount Yiwulü repeatedly, there was neither such a name "Beizhen" noted at that time nor any explicit record about sacrificial rituals for Mount

Yiwulü as early as the Warring States to Han Dynasty. The establishment of sacrifice for the five sacred peaks and four waterways was known to be formalized as a conventional state ritual in 61 BCE (Jia 2021, pp. 7–8), but historical records indicated that the stronghold's sacrifice was only formally included in the state ritual system of sacrifice later in the Sui and Tang dynasties (581–618; Wei 1973, 7.140; Jia 2021, p. 9). However, this does not undermine the importance of Mount Yiwulü before the Sui Dynasty.

The earliest mentions of the mountain range of the northern stronghold were found in the *Weishu* 魏书 (History of Wei), which indicates that the Northern Wei 北魏 (386–534) rulers noticed this grand mountain located in the north. The *Weishu* records that Tuoba Jun 拓跋濬 (r. 452–465), the fifth emperor of Northern Wei, made a tour to the east in the year 460 during which the northern stronghold was the second stop of his trip. The emperor first went to Qianshan桥山 (present-day Quwo, Shanxi) to worship the Yellow Emperor, and after Mount Yiwulü in western Liaoning, he returned to Shanxi to the northern sacred peak Mount Heng 北岳恆山, another sacrificial site. Since Mount Yiwulü was not in the territory of the Northern Wei at that time, Tuoba Jun performed a distant sacrificial ritual in western Liaoning to Mount Yiwulü (Wei 1974, 108a.2739). This was the earliest literary record of sacrifice to Mount Yiwulü. The Northern Wei Emperor's personal visit to the border region to perform the mountain sacrifice was an indication that during the Northern and Southern dynasties, the northern minority regimes were looking to expand their political influences to the northeastern area, in addition to the sacrificial sites of the western and northern sacred peaks established in the northern territory. Tuoba's tour was also an expression of the sovereignty of the northern minority regimes in northeastern Liaoning, although the state of Northern Wei did not have actual control there. Through the performance of state sacrifice, the Northern Wei regime tended to show they had the same or even higher legitimacy as the Southern regime, and they were the legitimate successor of the world under Heaven. Although Tuoba's eastern tour had not reached Mount Tai, the route taken had referred to all the routes of previous emperors' eastern tours for Mount Tai Sacrifice. As an ethnic Sienpi, Tuoba went so far as to model Emperor Wu of the Han dynasty's sacrifice to the Yellow Emperor, the ancestor of the Han nationality. These are all signs that Tuoba was asserting his political legitimacy using ways of the Han regime, and it was done to compete with the regimes in the south. Besides the Northern regime of Wei, the distant sacrifices to Mount Yiwulü had also happened occasionally in the southern region. After ascending to the throne, Xiao Yan蕭衍, the Emperor Wu of Liang dynasty 梁武帝 (r. 502–549), began to gather Confucian scholars to formulate the national ceremonies, which determined the alternating offering sacrifices to Heaven in the southern suburbs and to Earth in the northern suburbs every other year. Sacrifices in northern suburbs included the rituals to the five sacred peaks, four waterways and four seas, as well as Mount Yi 沂山, Mount Huo 霍山, and Mount Yiwulü, among which the four strongholds had not been formalized in the national ritual system of sacrifice (Wei 1973, 6.108). Apparently, due to the constraints of the military confrontation between the north and south, sacrifices to mountains and waters which are located outside the border could only be performed from a distance. Emperor Wu of Liang thus reintegrated the state rituals and made numerous mountains and waters in the northern region the objects of sacrifice as the Southern court's formal statement to legitimize their claim over the northern regions. Despite the conflicts, both rulers of the northern and southern regimes regarded Mount Yiwulü highly, although at times, neither of them had control over this area. The emphasis placed on Mount Yiwulü by both regimes had demonstrated their common recognition of the universal system under Heaven conceived in the *Zhouli*.

Following the unification of the country, the Sui dynasty set about consolidating the state ritual system of sacrifice in accordance with the perception of the Northern and Southern dynasties—the practice of performing state rituals based on the contents of the *Zhouli* was an important basis for the regimes to establish their political legitimacy during the previous dynasties. One of the more important initiatives was that the Sui incorporated the four strongholds' sacrifices into the state ritual system for the first time.

In addition, the Sui also established temples on each of these stronghold mountains (Wei 1973, 7.140). This system continued throughout the successive dynasties. In the early years of the Northern Song dynasty (960–1127), the official court added the Central Stronghold (Zhongzhen 中鎮, i.e., Mount Huo) to the original four strongholds system and since then, the sacrifice system of the five strongholds had been formally formed. Additionally, the imperial court granted duke titles to the five sacred peaks, five strongholds, four seas, and four waterways, among which Mount Yiwulü was granted the title "Duke of Grand Peace" (Guangning Gong 廣寧公) (Toqto'a 1977, 102.2488; Jia 2021, p. 10). However, Mount Yiwulü at that time was in the territory of the Liao dynasty, so the Northern Stronghold Temple of Mount Yiwulü and several other temples, such as the North Sea Temple and West Sea Temple, were not within the sphere of control of the Northern Song dynasty. As a result, the rulers of Northern Song relocated the sacrifice site from the Northern Stronghold to the Northern Sacred Peak (Beiyue 北嶽) Temple in Dingzhou 定州 instead (Jia 2021, pp. 10–11). No record of sacrifice to the northern stronghold by the Liao court was found, which makes it impossible to trace the history of the Northern Stronghold Temple, now located in Beizhen City, Liaoning Province, back to the Sui and Tang Dynasties.

There is no direct evidence in existing historical documents indicating the Liao regime followed the state ritual system of sacrifice from the Chinese central regime, but Mount Yiwulü was more than just a site "defending the north" to the Liao because the northeastern region was the birthplace of the Khitan. After the Khitan had established the Liao state, the imperial family designated Mount Yiwulü as one of the sites for the imperial mausoleums. Among the five imperial mausoleums of the Liao dynasty, Xian mausoleums (Xianling 顯陵), Qian mausoleums (Qianling 乾陵) are located in the Mount Yiwulü area, where buried four of the nine emperors of Liao, as well as several empresses and princes (Yu and Bai 2020, pp. 27–33; Liaoning Sheng Wenwu Kaogu Yanjiusuo 2016, pp. 34–54). From this point of view, Mount Yiwulü had indeed unparalleled importance to the Liao imperial family. The protection of the area around the Liao imperial mausoleums in Mount Yiwulü continued in the Jin dynasty. For example, in 1129, Wanyan Sheng 完顔晟, Emperor Taizong of Jin 金太宗 (r. 1123–1135), banned woodcutting around the Liao mausoleums to protect the areas around it (Toqto'a 1975, 3.60). However, because of the operation of the Liao regime in the area of Mount Yiwulü, the ritual system of sacrifice for the Northern Stronghold since the Sui and Tang Dynasties was not continued for over two hundred years, so the temples established during the Sui and Tang had also disappeared.

Unlike the Liao dynasty, the Jin dynasty, as a Jurchen regime, formally adopted the state ritual system of sacrifice from the central Chinese kingdom. Wanyan Yong 完顔雍, Emperor Shizong of Jin 金世宗 (r. 1161–1189), was the first Jin emperor to follow this system. In the sixth month of 1164, he resumed the sacrifice rituals to the five sacred peaks, five strongholds, four seas, and four waterways (Toqto'a 1975, 6.134; 34.810). It is generally believed that the construction of the Northern Stronghold Temple, now located in Beizhen City, Liaoning Province, was built from this time. Like the Song dynasty, the Jin government sent officials to visit mountains and waters within the country to perform sacrifice rituals, such as the sacrificial ritual of Mount Yiwulü in Guangning 廣寧, and conducted distant sacrifices of mountains and wasters outside the country's borders. The Jin also sacrificed to the earth spirit in the suburb of the capital, as well as set up spirit tablets for mountains and waters (Toqto'a 1975, 29.712). The Jin dynasty also followed the old system of the Tang and Song dynasty, granting duke titles to these mountains and waters. In 1190–1196, the Daoists' suggestion to follow the example of the Northern Song and confer the mountain and water spirits as Kings was adopted (Toqto'a 1977, 102.2488). Mount Yiwulü was then given the title "King of Grand Peace" (Guangning Wang 廣寧王) (Toqto'a 1975, 34.810).

The above materials from the pre-Qin to Song and Jin periods showed the earliest documentation of Mount Yiwulü as one of the four strongholds (later became the five strongholds). However, for hundreds of years after the Han dynasty, the mountain was not deemed eligible to enter the state ritual system of sacrifice until the Sui dynasty. The minority regimes' special attitude towards Mount Yiwulü during this period was

revealed, on the other hand, because this region was under the control of the northern regimes for many years, and the sacrifice to this mountain was an important aspect in determining the system of a unified common world under Heaven mentioned in the *Zhouli*, which was a significant basis for declaring the legitimacy of their own regimes. Later, with the sinicization of the minority regimes in the north and the urgent desire to enter the Central Plains, the requirements for contending the political orthodoxy also increased. Through the Yuan and Qing Dynasties, this special attitude was further reinforced and contributed to a new connotation of Mount Yiwulü. In comparison, the Chinese regimes regarded Mount Yiwulü as a part of the entire sacrificial system and did not give it special treatment.

### 3. The National Root Place of Vitality: The Sacrifice of the Northern Stronghold in the Yuan Dynasty

According to historical documents, most of the references to the northern stronghold sacrifice are mentioned in conjunction with other mountains and waters, and there is not much said about the stronghold's particularity. Fortunately, more than fifty historical stele inscriptions in the Northern Stronghold Temple provide an important glimpse into the history of the sacrifice to the northern stronghold and its historical position, particularly after the Song dynasty. The earliest surviving stele inscription in the Northern Stronghold Temple is the "Monument of the Holy Commandment", which was erected in 1298. The inscription records the history that the five strongholds were granted the King title by Borjigin Temür 鐵穆爾, Emperor Chengzong of Yuan 元成宗 (r. 1295–1307). This edict was also made for stone steles and sent to the other four strongholds. According to the inscription, Temür believed that all the previous emperors before the Yuan dynasty had ennobled the five sacred peaks and four waterways (Song 1976, 72.1780; 76.1900), but did not perform sacrifices for the five strongholds. According to *Yuanshi*, 元史 (the History of Yuan Dynasty), Yuan Emperors did not go to the sacrificial site personally, but sent high-ranking officials accompanied by Han Confucian scholars and Taoist priests. This tradition began in 1261. In 1291, Kublai Khan 忽必烈 (r. 1260–1294) conferred the title of Emperor for the five sacred peaks and the title of King for the four waterways and four Seas, but did not confer the titles of the five strongholds (Song 1976, 72.1780; 76.1900). Therefore, Temür especially granted the title of king to the spirits of the five strongholds, and prayed that these strongholds could fulfill their duties of pacifying the people and nurturing the universe:

> The five sacred peaks and four waterways had already been granted titles by emperors, but the five strongholds' sacrifice alone had not been recognized. This was indeed not meant to be for the worship of divinities . . . The emperor ordered relevant official departments seasonally perform sacrifices to the five strongholds together with the five sacred peaks and four waterways and formalized the standard, thus this edict was issued to inform the public. 五嶽四瀆，先朝已嘗加封，唯五鎮之祀未舉，殆非敬恭明神之義。 . . . . . . 仍敕有司歲時與嶽瀆同祀，著為定式，故茲詔示，想宜知。 (Borjigin 1298/2002, pp. 48–49)

The ritual system of sacrifice for the five strongholds was first proposed by the Northern Song and was continued in the Jin dynasty. However, as neither Northern Song nor Jin had control over the entire empire, the sacrifices to the five strongholds were never practiced as a matter of fact. Temür thus became the first emperor to successfully perform the investiture and ritual to the five strongholds as he desired to demonstrate that the Yuan had contributed to the unification of the country.

Since then, the Yuan government had sent ministers to the Northern Stronghold Temple on many occasions to make sacrifices. For example, the stele inscriptions show records of the sacrifice rituals performed in the temple in the year 1313, 1317, 1339, 1342, 1343, 1346, 1347, 1348, and 1357 (Wang 2002, pp. 51–55, 217–23; Yu 2009, pp. 9–60). The contents of these steles are primarily about worshiping the mountain spirit, praying for good harvest, and blessing the country with peace and prosperity. These continuous

sacrifices to Mount Yiwulü ended in the late Yuan when the official reverence for Mount Yiwulü began to differ from other strongholds.

In 1339, the sacrificial officials believed that Mount Yiwulü was supposedly the stronghold mountain of Youzhou 幽州, and the survival of the state was dependent on this mountain (Li 1339/1983, 255.5574). This is because Youzhou was not only a crucial military town for a long time, but also was the capital of the Yuan dynasty, so Mount Yiwulü located in Youzhou was related to the foundation of the country. The phenomenon of sacred sites changing in geographical and hierarchical significance is not unique. The Southern Sacred Peak had once been regarded as the most important sacred peak instead of the Eastern Sacred Peak (Robson 2009, pp. 57–89). Moreover, the stele inscription of 1346 ("Yuxiang daisi ji" 御香代祀記 (Record of Imperial Incense-offering and Sacrifice on Behalf of Emperor)) clearly states that Mount Yiwulü was the place where the root vitality of the nation lied, and had a higher status than the other strongholds:

> Until our grand Yuan dynasty, the Northern Stronghold was conferred with the noble title of Faithful Virtue King. Emperors held solemn sacrifices grander than all previous dynasties because the stronghold of Youzhou was closely related to the capital's safety. The Northern Stronghold is the root place of our vital national force, which is more [three characters missing] than the other strongholds. 迨我皇元，崇秩貞德王號，列聖嚴禋，比之累代褒封欽重者，實主鎮幽州，皇都京畿係焉。乃我國家根本元氣之地，較之異方山鎮，尤為□□□焉。 (Zhang 1346/2002, pp. 219–20)

This is the first description in the available sources that made the status of the Northern Stronghold the most superior among the five strongholds, whereas previously, it was generally considered that the Eastern Stronghold Mount Yi was at the top of the hierarchy. These inscriptions were written by Han officials, and they used "I think" in writing to express their stance. These Han officials claimed the status of the Northern Stronghold by referring to both the Confucian canons and the Yuan emperor's granting. The emphasis on their attitude and feeling highlighted the importance of the Han officials' recognition of the political legitimacy of the Yuan dynasty. The reason for this change in the status of Mount Yiwulü is closely related to its geographical location. Mount Yiwulü was the gateway to the Yuan territory and one of the natural barriers that guarded the capital. In the stele inscription of 1347 ("Yuxiang daisi ji" 御香代祀記 (Record of Imperial Incense-offering and Sacrifice on Behalf of Emperor)), the Northern Stronghold was described as the birthplace of the Yuan dynasty as they were both in the north (Zhang 1347/2002, pp. 220–21). In the stele inscription of 1357 ("Daisi zhibei" 代祀之碑 (Monument of Sacrifice on Behalf of Emperor)), Mount Yiwulü was considered the cause of the exquisite change between heaven and earth, which has the meaning that the imperial power of the Yuan originated here (Yang 1357/2002, p. 223). Although the area of Mount Yiwulü was not the birthplace of Mongolia and was not closely related to the Yuan regime, it was the only stronghold that was associated with the northern minority regimes. As discussed above, from the Northern Wei, Mount Yiwulü began to be regarded highly by the ruling class. The Khitan royal family chose it as the site of the imperial mausoleums, and the Jin established a temple there for the first time.

In the last years of the Yuan dynasty, the rulers' attitude towards Mount Yiwulü became more respectful. However, at that time, many incidents indicate that the country's solid regime was beginning to disintegrate as uprisings constantly took place in the south. On the one hand, the Yuan government's reaffirmation of the importance of the Northern Stronghold served to emphasize absolute control over the northern region, especially around the state capital (i.e., the military town Youzhou). As nomadic people in the north, the Yuan had almost swept away all threats from the north, thus the place was also the region where their power was most secure. On the other hand, it can also be implied that the Yuan government may have been considering various ways to strengthen the legitimacy of its domination in these turbulent times, one of which might be the secured

north represented by Mount Yiwulü, which stood for the cornerstone of the regime's power and the foundation that the regime had to secure and rely on.

## 4. The Frontier, Military Battlefront, and Head of the Five Strongholds: The Sacrifice of the Northern Stronghold in the Ming Dynasty

The exalted status of Mount Yiwulü ceased to exist with the establishment of the Ming dynasty (1368–1644). In the nearly three hundred years of the Ming, the Northern Stronghold once again became the frontline of the confrontation between the Chinese regime and the northern nomadic peoples, especially with the newly emerging Jurchen tribes (subsequent Manchurians). In addition to performing the same religious functions as other strongholds, the military status of the Northern Stronghold was given special emphasis throughout the Ming dynasty.

In the third year of Hongwu 洪武 (1370), Zhu Yuanzhang 朱元璋, Emperor Taizu of Ming 明太祖 (r. 1368–1398), reformed the state religious system by removing all the "blasphemous" human titles of the five sacred peaks, five strongholds, four seas, and four waterways, and added divine titles to them to show reverence. Since then, the specifications of the sacrifices to state mountains and waters had been raised to an unprecedented level. In the process, the Northern Stronghold no longer received special religious treatment as it had previously by the northern minority regimes but returned to being treated more equally as one of the five strongholds. However, with the Ming dynasty's northern expedition against the Mongolian remnants and the subsequent moving of the capital to Beijing to defend the country against the Mongolian invasion from the north, Mount Yiwulü, with its natural role as a military barrier, came into the view of the central authorities again. As the only connecting passage between the capital and Liaodong, and the last barrier outside the Shanhaiguan 山海關, Guangning, where the Northern Stronghold was located, was vital to the security of the capital and even the whole country. In view of this important military position, the Liaodong region withdrew all its original administrative organs from the beginning of the Ming and became a military organization, the Liaodong Commanders' Department (Liaodong Duzhihuishi Si 遼東都指揮使司), and Guangning became a heavily guarded place. It was also for this reason that all the sacrificial officials recorded in the existing Ming dynasty stele inscriptions from the Northern Stronghold Temple were all in important military positions.

Because of its importance after being tested by these historical incidents, the Northern Stronghold received special attention from the Ming court. Since the nineteenth year of Yongle 永樂 (1421) period, the official renovation of the Northern Stronghold Temple began and continued throughout the dynasty. As many as the six documented renovation programs have been made are enough to show how importantly the Ming government treated this only place of state sacrifice with military functions (Zhu 1421/2002, pp. 59–60; Zhang 1495/2002, pp. 226–27; Huo 1509/2002, pp. 228–29; 1606/2002, pp. 242–43). Thus, as mentioned in the "Beizhenmiao chongxiu ji" 北鎮廟重修記 (Record of the Restoration of Northern Stronghold Temple) in 1495, the Northern Stronghold was not only revered as a sacred mountain but also a reliable fortress at the border for defensive purposes:

> The Northern Stronghold ranks the first among the stronghold mountains in the national sacrifice system, forever stabilizing the eastern land and benefiting the people living in the border areas, which is the same in merit as the five sacred mountains and four waterways. 北鎮禮秩居他鎮之首，永奠東土，御我邊疆，利我邊民，與五嶽四瀆同功。 (Zhang 1495/2002, pp. 226–27)

This inscription also recorded that it was the first time the Ming dynasty had explicitly identified the status of the Northern Stronghold as the most important mountain to guard the northeastern territory, as it was the head of the five strongholds. The "Chongxiu Beizhenmiao beiji" 重修北鎮廟碑記 (Reconstruction of Northern Stronghold Temple Monument) in 1509 also expressed that the Northern Stronghold was related to the peace and stability of the whole country (Huo 1509/2002, pp. 228–29). In summary, it appears that

almost all the surviving Ming dynasty stele inscriptions referred to the importance of the military status of the Northern Stronghold.

Throughout the Ming dynasty, the area of Guangning, where the Northern Stronghold was located, was closely associated with the resistance to the northern minorities, first Mongolian and then Manchurian. It is apt to say that the survival of the entire country was related to the situation of Guangning, which, to a great extent, determined the national fate of the Ming and Later Jin Dynasties. For Manchuria, military control of the Guangning area was decisive for its access to Shanhaiguan and the eventual establishment of the Qing dynasty (Twitchett and Fairbank 2008, pp. 41–49, 52–57).

**5. The Origin of Manchuria: The Sacrifice of the Northern Stronghold in the Qing Dynasty**

The Qing dynasty's treatment of the Northern Stronghold had achieved the most glorious time in the Northern Stronghold's history. In the previous dynasties, emperors rarely went in person to the Northern Stronghold to perform the sacrifices, while the Liao dynasty had only chosen Mount Yiwulü as the imperial tomb. During the Qing dynasty, five emperors personally went to the Northern Stronghold eleven times in total to sacrifice the spirit of Mount Yiwulü. They performed national rituals there and repeatedly inscribed inscriptions and poems to express their reverence. It is precisely because of the special attention of the Qing dynasty imperial family that the Northern Stronghold Temple has been preserved to this day, becoming the only stronghold temple that preserved the Ming and Qing architecture. The reasons why the Qing emperors valued the Northern Stronghold are obvious. The Northern Stronghold was both one of the traditional sites of the state rituals of sacrifice for mountains and waters and the site where the ancestors thrived and were buried. As early as the Wanli 萬曆 period of the Ming dynasty (1573–1620), the ancestors of Aisin-Gioro Nurhachi 努爾哈赤 (r. 1616–1626) had already established a family mausoleum outside Hetu Ala 赫圖阿拉 (in present-day Xinbin County, Fushun City, Liaoning Province), which later became the Yong Mausoleum 永陵. From the third year of Tiancong 天聰 (1629) to the eighth year of Shunzhi 順治 (1651), Nurhachi's Fu Mausoleum 福陵 and Huang Taiji's 皇太極 (Aisin-Gioro Hong Taiji, r. 1626–1643) Zhao Mausoleum 昭陵 were inaugurated in Shengjing 盛京 (present-day Shenyang, Liaoning). The city of Hetu Ala and Shengjing served as the base camps of the Manchurian and enshrined their ancestors. For this reason, after the establishment of the Qing dynasty, the emperors had the habit of the eastern tour to worship their ancestors. Guangning, where the Northern Stronghold was located, was a designated stopping point for the Qing emperors on their way to sacrifice to their ancestors. Next door to the temple was the Guangning Palace, built for the emperor's temporary rest. According to historical records, there were five emperors of the Qing dynasty who personally visited the East to worship their ancestors. These tours were led by Emperor Kangxi 康熙帝 (Aisin-Gioro Xuanye 愛新覺羅·玄燁, r. 1662–1722) three times (1671, 1682, 1698), Emperor Yongzheng 雍正帝 (Aisin-Gioro Yinzhen 愛新覺羅·胤禛, r. 1723–1735) once (1721), Emperor Qianlong 乾隆帝 (Aisin-Gioro Hongli 愛新覺羅·弘曆, r. 1736–1795) four times (1743, 1754, 1778, 1783), Emperor Jiaqing 嘉慶帝 (Aisin-Gioro Yongyan 愛新覺羅·琰, r. 1796–1820) twice (1805, 1818) and Emperor Daoguang 道光帝 (Aisin-Gioro Minning 愛新覺羅·旻寧, r. 1821–1850) once (1829). Some of their activities are also recorded on the stele inscriptions of the Northern Stronghold Temple.The "Yuji zhuwen" 御祭祝文 (Imperial Sacrifice Blessing Stele) in 1682 was written on the way of Kangxi's second ancestor worship, for which Kangxi sent his close courtiers to perform sacrificial rituals to the Northern Stronghold. With the successful suppression of a nearly decade-long rebellion, the country had regained peace, and the power of Emperor Kangxi was secured, this tour was significant. This inscription argues that Mount Yiwulü was not only the place where the Manchurian race was born and emerged, but also the place where the royal energy gathered. It was believed that with the blessing of the spirit of Mount Yiwulü that Emperor Kangxi was able to quell the rebellion and restore stability of the country:

The god stands majestically in the land of Yingzhou and the coast of Liaohai, which is the birthplace and foundation of our ancestors and a place filled with the kingly energy (qi). Blessed by the god, I put down the rebellions. 維神傑峙營州，雄幡遼海，發祥兆跡，王氣攸鍾。朕祇承神祐，疆宇蕩平。 (Aisin-Gioro 1682/2002, p. 245)

This comment not only regarded Mount Yiwulü as the origin of the Manchurian race and Qing regime, but also as a symbol and source of imperial power, which further elevated the status of the Northern Stronghold. This evaluation continued until the end of the Qing dynasty, and the Northern Stronghold was praised by all the successive Qing emperors after Kangxi.

"Xinjian Beizhen Yiwulü shan zunshen bange xu" 新建北鎮醫巫閭山尊神板閣序 (Preface of Newly Built Sacrifice Pavilion of the Northern Stronghold Mount Yiwulü) in 1690 mentions the significance of Fengyi 丰邑 and Haoyi 鎬邑, the birthplaces of Zhou dynasty, to allude to the significance of Mount Yiwulü to Qing dynasty (Huang 1690/2002, pp. 245–46). Emperor Kangxi passed through Mount Yiwulü several times on his eastern tours, and often viewed beautiful clouds rising from the emerald green peaks, which are seemingly connected to Heaven. From this sight, the belief was formed that the cloud of Mount Yiwulü was the imperial energy descending from heaven, Mount Yiwulü was the place favored by heaven's mandate, and the Manchurians, who originated in this area, had gained power in accordance with it (Aisin-Gioro 1708/2002, p. 249). Therefore, in 1703, Kangxi specially wrote a four-character plaque of "Lush and Auspicious Energy" (*Yucong jiaqi* 鬱蔥佳氣) for the Northern Stronghold Temple ("Yuji zhuwen bei" 御祭祝文碑 (Imperial Sacrifice Blessing Monument)) (Aisin-Gioro 1703/2002, pp. 247–48). Later, Kangxi ordered officials to reconstruct the temple with meticulous care, which took two years and four months (1706–1708). The "Yuzhi beiwen" 御製碑文 (Imperial Monument) in 1708, written in bilingual Manchu and Chinese by Kangxi himself, relates that Mount Yiwulü not only guarded the imperial spirit of the Qing dynasty, but also was an auxiliary and fence to the capital that consolidated the foundation of the imperial family for ten thousand years (Aisin-Gioro 1708/2002, p. 249). Kangxi was the first Qing emperor to openly express his admiration for the Northern Stronghold. In addition to emphasizing the political and military functions of Beizhen as well as the Yuan and Ming dynasties, emperor Kangxi's praise of the Northern Stronghold was more based on national emotions. More notably, in comparing the Northern Stronghold to Feng and Hao, the birthplace of the Zhou dynasty, he also secretly compared himself and his ancestors to sage kings like the Kings of Zhou. Using the Northern Stronghold as the link, Emperor Kangxi skillfully connected the origin of his own nation with the legitimacy of the political power and compared it with the Zhou dynasty, the ideal blueprint of the Han political power in the Central Plains. In this way, to a certain extent, he was able to win the political identification of the Han literati, which proved his brilliant political skills. Qian Shixun 錢世勳, the governor of Guangning, was one of the Han literati who highly praised Kangxi as a wise and brilliant emperor. He wrote several inscriptions to pray for the emperor's long life. He also hoped people would cherish the emperor with feelings of affection and gratitude (Qian 1712/2002, pp. 250–51; Qian 1715/2002, pp. 251–52). Kangxi's reverence for the Northern Stronghold Temple was widely spread and received praise from people in the local area. He exemplified the attitude toward the Northern Stronghold Temple to be carried on by later emperors of the dynasty.

Emperor Yongzheng's only ancestral worship took place in 1721 before he ascended the throne. To celebrate the sixtieth anniversary of Kangxi's reign, Yin Zhen 胤禛, the crown prince, was ordered to replace Kangxi on an eastern tour to worship ancestors, and this deed was recorded in "Yuzhi beiwen" 御製碑文 (Imperial Monument) in 1727 written in both Manchu and Chinese by Yongzheng himself. Yongzheng also made the decision to sponsor the renovation of the temple during this trip. Besides, after Yongzheng ascended the throne, he immediately ordered officials to spend four years repairing the temple again. The two repairs within just a few years show Yongzheng's deep feelings

for the Northern Stronghold Temple. The inscription, written by Yongzheng himself, first summarized Kangxi's reverence for Mount Yiwulü for more than sixty years, then further confirmed that Mount Yiwulü was the birthplace of his people, and finally emphasized that the legitimacy of the Qing dynasty's regime came from the gathering of the "imperial energy" of Mount Yiwulü and the support of the mountain spirit. Emperor Yongzheng fully inherited and carried on Emperor Kangxi's reverence for the Northern Stronghold:

> Mount Yiwulü is indeed the Northern Stronghold close to Xingjing [Hetu Ala], which guards the nearby areas. 醫巫閭山，實為北鎮，近接興京，翊衛關輔。
>
> Our ancestors are established in the east of Shanhaiguan, like Feng and Qi of the Zhou dynasty, it is a region filled with kingly energy (qi). This place is blessed by the god with frequent miraculous happenings and laid the foundation for the great achievements of our nation permanently. 我祖宗發祥關右，豐、岐重地，王氣所鍾。惟神實為擁護，永奠鴻基，靈績頻昭。
>
> With the intention of proper deference, I revere the deities from morning to night. Therefore, the god sent blessings, continuously revealing its great achievements. 朕懷允愜展敬之念，夙夜加虔。神其宏敷蕃祉，益顯豐功。 (Aisin-Gioro 1727/2002, p. 252)

Yongzheng constantly stressed that he sacrificed on behalf of his father as the prince in Kangxi's later years, which may be a declaration of the legitimacy of his succession after the fierce struggle for succession.

Although Emperor Qianlong did not leave any sacrifice inscriptions in the Northern Stronghold Temple during his four eastern ancestral tours, he did leave more than a dozen poems related to Mount Yiwulü and poetry stelae written by himself. Almost all the surviving poetic inscriptions in the Northern Stronghold Temple were works of Qianlong (except for one poem written in the shade of an inscription by Emperor Daoguang). Emperor Kangxi and Yongzheng also left poems written on Mount Yiwulü, though not as many as Qianlong's. Emperor Kangxi's work "Guo Guangning wang Yiwulü shan" 過廣寧望醫巫閭山 (Looking from afar at Mount Yiwulü when passing Guangning) expresses his desire to pass through Guangning and look at Mount Yiwulü from afar, and his desire to climb this mountain to cultivate himself. Emperor Yongzheng's "Wang Yiwulü shan" 望醫巫閭山 (Looking from afar at Mount Yiwulü) praises the ancestors' inheritance in Mount Yiwulü and emphasizes the legitimacy of the imperial power of the Qing, which he attributes to this mountain. Emperor Qianlong's poems, on the other hand, took on a more multifaceted appearance and brought out sentiments to the extreme.

Emperor Qianlong stayed in Mount Yiwulü every time he was on his eastern tours to worship the ancestors and left his own poems and erected monuments to commemorate them. Among the surviving inscriptions, there was one piece written in 1743, seven pieces in 1754, nine pieces in 1778, and eight pieces in 1783 (Wang 2002, pp. 450–51, 453–57; Yu 2009, pp. 158–95). In addition to some of these pieces depicting scenes of sacrifice, wishes for blessing, and the exploits of ancestors, most of them express personal feelings of the emperor. Some of these poems are about lingering on the beautiful scenes of Mount Yiwulü (Qianlong called them the "seven scenes") (Aisin-Gioro 1754c/2002, p. 451). In some of his works, Qianlong alludes to the emperors of the Liao dynasty who lived in seclusion but aimed at the world to convey his same aspirations (Aisin-Gioro 1754a/2002, p. 450; Aisin-Gioro 1778/2002, p. 455). In others, he describes the peaceful and tranquil village life in Mount Yiwulü (Aisin-Gioro 1743/2002, p. 450; Aisin-Gioro 1783/2002, p. 456), nostalgia upon imagining the ancient wars that took place in this area, reflection on the lessons from the fall of the Ming dynasty, and affirmation of the virtues and merits of the Qing dynasty (Aisin-Gioro 1754b/2002, pp. 450–51). Overall, compared to the Northern Stronghold described by the previous emperors, these poems present a broader scene, deeper historical reflections, more realistic images of life, and more personal feelings. Qianlong was the only emperor who expressed his personal emotions and feelings through the Northern Stronghold. Apart from the personality factors of emperor Qianlong, his tendency towards

the Northern Stronghold sacrificial rites also came from the confidence and relaxation brought by the stability of the regime and the prosperity of the country. In Qianlong's time, the problem of regime legitimacy had been basically resolved, and he no longer needed to elaborate on it. As a result, Emperor Qianlong turned his attention away from politics and focused instead on the local landscape and people's livelihood, as well as his personal feelings.

The last restoration of the Northern Stronghold Temple in Chinese imperial history took place in the eighteenth year of the Guangxu 光緒 (1892) period, and was recorded in a lengthy stele inscription. This "Chixiu Beizhenmiao bei" 敕修北鎮廟碑 (Monument of Imperial Reconstruction of the Northern Stronghold Temple) inscription detailed the historical evolution of Mount Yiwulü for more than 2000 years and the reconstruction of this temple in previous dynasties. This text argues that the spiritual vein of Mount Yiwulü is connected to Mount Changbai 長白山, the place of origin of the Manchus and the nearest natural barrier to Hetu Ala and Shengjing, which was crucial to the Manchus. The fact that all the emperors highly regarded and revered Mount Yiwulü demonstrated the importance of remembering one's origin in the Qing dynasty (Chen and Xu 1892/2002, pp. 289–91). In fact, the Qing dynasty regarded Mount Changbai as the original birthplace of its race, and Mount Yiwulü as the place where the emperor's foundation began to flourish and prosper. Therefore, the Qing dynasty had a tradition of performing sacrifices to Mount Changbai and Mount Yiwulü together (Zhao 1977, 83.2522). After Emperor Guangxu ascended the throne, he also added the divine titles of Mount Changbai and Mount Yiwulü, the former as "Protect the People" (Baomin 保民) and the latter as "Accurate Response" (Lingying 靈應) (Zhao 1977, 83.2523).

To sum up, the Qing dynasty promoted the Northern Stronghold for many reasons. The first is that the Qing dynasty, as an ethnic minority regime, was in urgent need of inheriting the state ritual system of sacrifice from the Chinese regimes like the Jin dynasty and the Yuan dynasty to confirm the legitimacy of its own rule. This ritual system concretized the source of legitimacy for the regime through rituals, entertainment, and prayers to gain the support of the gods in order to obtain good harvest, which were all representations of the mandate of heaven. Secondly, it comes from the decisive military victory of the Manchurians over the Ming dynasty. Guangning was at the border of conflicts between the Later Jin and Ming armies, and the two sides fought for decades. The Manchu army occupied Guangning and then soon overthrew the Ming dynasty. At the same time, Guangning was also the only access to the Liaodong region from Beijing, which showed the importance of its military status. Thirdly, the Manchurians regarded Mount Yiwulü as the place where the Qing dynasty was founded. The Manchu Later Jin regime built its capital in the city of Hetu Ala and Shengjing, both in the east of Mount Yiwulü. The Northern Stronghold was a natural barrier to the political center of the Later Jin, which gave the Manchu regime a respite and a chance to grow under the military pressure of the Ming dynasty. The Qing government combined the special significance of Mount Yiwulü to the Manchus with the state ritual system of sacrifice, which developed the understanding that Mount Yiwulü was the place where the heavenly mandate was given and kingly energy was gathered, further strengthening the legitimacy of its regime.

## 6. Conclusions

This article uses extant stele inscriptions preserved in the Northern Stronghold Temple to examine the history of the Northern Stronghold sacrifice, focusing on the special attitudes of the northern minority regimes to Mount Yiwulü. While the sacrifices of the five strongholds were incorporated into the traditional state ritual system of sacrifice as late as the early Song dynasty, the Northern Stronghold sacrifices were valued by the ruling class as early as the Southern and Northern Dynasties. Moreover, the ethnic minority regimes in the north regarded the Northern Stronghold higher than the Chinese central regimes. The Northern Wei, Liao, Jin, Yuan, and Qing dynasties all gave special treatment to the

Northern Stronghold sacrifice. The Yuan and Qing even regarded the Northern Stronghold as the head of the five strongholds. This positive attitude was formed for several reasons.

Firstly, Mount Yiwulü was associated with the origins and prosperity of many northern peoples, such as the Liao of the Khitan and the Qing of the Manchurians. Even though people of the Jurchen Jin dynasty and the Mongol Yuan dynasty did not regard it as the birthplace of their nation, they still highly worshipped it. Liao and Qing had a closer connection with the Northern Stronghold, so they had the highest regard for this mountain. While the rulers of the Liao dynasty built the imperial mausoleums there, the rulers of the Qing dynasty not only considered Mount Yiwulü as the source of its imperial power, five emperors also personally went to the Northern Stronghold Temple to perform sacrifices on the way back to Liaodong to worship their ancestors.

Secondly, the geographical location and military role were also realistic reasons that the Northern Stronghold was important. Youzhou, where the Northern Stronghold is located, has been an essential fortress in the north since ancient times. From the Jin dynasty onwards, many dynasties set their capitals in Beijing. Due to the obstruction of the Mongolian Plateau, the only link between Beijing and the north, especially the north-east, was the Guangning area. For this reason, since the Yuan dynasty, Mount Yiwulü had been regarded as a natural barrier to protect Beijing. The Later Jin, founded by the Manchurians, treated Mount Yiwulü as a natural obstacle to Shengjing 盛京. While in the late Ming dynasty, the military confrontation became intensified between the Ming government and the Manchurians, thus the stability of the Guangning area would decide the survival of the regimes. Naturally, Mount Yiwulü, which secured the tranquility of this place, had also been highly valued by all parties.

Finally, the existing research on the Northern Stronghold tends to pay more attention to the study of its religious rituals and cultural background in a specific dynasty. However, the Northern Stronghold's presentation of a systematic opposition between ethnic groups and how ethnic minority regimes skillfully adopted a national sacrificial system that was originally against them by establishing a connection between their nationalities and the sacrificial subject, like the Northern Stronghold, are often neglected by scholars. The system of a unified common world under Heaven mentioned in the *Zhouli* is one of the most important sources of political legitimacy of the Han dynastic regime. This system was initially designed to distinguish the Han Chinese from the barbarians, so one of its core concepts is "identity", which regarded the ethnic minorities in the frontier areas as enemies, rebels, or people to be pacified. In this system, stronghold mountains serve to guard the border and resist foreign nationalities, and because they maintain the Han regime's political legitimacy, the stronghold mountains' embodiment of the conflict between the Han and other nationalities is particularly severe. Due to historical and geographical reasons, this conflict is further reified by the Northern Stronghold.

However, under the continual operation of the northern minorities, the Northern Stronghold transcended the national sacrificial system based on the Han nationality and its regime's political legitimacy. The reason for such operation is that the system established by the *Zhouli* already encapsulated a sense of completeness. Rather than creating a new system, integrating itself into the system that it was excluded from was a better choice. The core operational means of ethnic minority regimes is still "identity". The act of binding one's identity to the Northern Stronghold, one of the symbols in the *Zhouli*, not only helped to integrate itself into the system but also transformed oneself from the enemy to the protected.

In this process, the minority regime's own statements are very important, but more important is the attitude of the Han officials. Therefore, we can see that in the stele inscriptions of the Yuan dynasty, the officials sent by the emperor to preside over the sacrifices were all Mongolians, while those who wrote the inscriptions to praise the virtues were all Han officials. In the Qing dynasty, the emperor wrote personally for the inscriptions. Under this influence, many local officials and scholars wrote inscriptions for the Beizheng Temple to express their support for the reestablished status of the Northern Stronghold in

the Qing dynasty. The actions and statements of Han officials are critical in justifying the legitimacy of these minority regimes, so they always occupied a significant place in the stele inscriptions.

Ming dynasty's attitude towards the Northern Stronghold reflected the Han regime's strong objection to the minority regime's manipulation of the identity of the Northern Stronghold. The Han regime once again stressed the military status of the Northern Stronghold, the purpose of which was to exclude ethnic minorities as enemies from the national sacrificial system established in the *Zhouli*. In this sense, the Northern Stronghold sacrificial system not only embodied the religious and cultural meaning in the other stronghold mountains but also added a layer of complexity to fighting for the power of speech between the Han and ethnic minority regimes. This is a distinctive aspect of the Northern Stronghold in the national sacrificial system, which provided a new theoretical dimension for the interpretation of the national sacrificial system.

Due to the mutual and combined effects of the above reasons, the Northern Stronghold was far more prominent than other strongholds as regarded among the minority regimes and was revered and cared for in particular manners by them. While these are phenomena that are hardly included in official history, the stele inscriptions preserved in the Northern Stronghold Temple serve to fill the gap in our knowledge, presenting the historical rise and fall of the Northern Stronghold over the past thousand years.

**Author Contributions:** Conceptualization, C.H.; Data curation, C.H.; Formal analysis, C.H.; Investigation, C.H. and S.C.; Methodology, C.H.; Resources, C.H.; Supervision, C.H.; Validation, C.H. and S.C.; Writing—original draft, C.H.; Writing—review & editing, C.H. and S.C. All authors have read and agreed to the published version of the manuscript.

**Funding:** This research received no external funding.

**Institutional Review Board Statement:** Not applicable.

**Informed Consent Statement:** Not applicable.

**Data Availability Statement:** Not applicable.

**Conflicts of Interest:** The authors declare no conflict of interest.

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
