# Peer review of "The Northern Stronghold Sacrifice and the Political Legitimacy of Ethnic Minority Regimes in the Late Imperial China"

_religions, doi:10.3390/rel13040368_

Round 1

Reviewer 1 Report

The advantage of this paper lies in its full, and accurate materials. It is  readable and attractive, and well structured. However, the author needs to present more evidence to prove the point “how the ethnic minority regimes successfully utilized the traditional Chinese state sacrificial ritual to serve their religious purpose” (Line 29-30)

Author Response

In this paper, we mainly focus on the political practice and activities of the ethnic minority who treated the Northern Stronghold in traditional Chinese state sacrificial ritual as a source of political legitimacy for their regimes. Some religious activities are mentioned in this paper. The ethnic minority performed almost the same sacrificial ritual to the Northern Stronghold as the Han people did. The religious aspect is not the main topic of this paper, but the political aspect is. As a response, I delete the two terms “religious” and “cultural” in this sentence to improve the clarity of the abstract.

Reviewer 3 Report

Overall this is a very interesting article. Its only problem is numerous points at which the language is somewhat unclear. I have recommended some changes to improve the clarity of the article in the attached file.

Author Response

The reviewer recommended many changes to improve the clarity of this article. We have accepted all the suggestions.